# Effect of Defatting Method on Japanese Quince (*Chaenomeles japonica*) Fruit Seed Protein Isolate Technological Properties

**DOI:** 10.3390/foods14020234

**Published:** 2025-01-13

**Authors:** Danija Lazdiņa, Dalija Segliņa, Zaiga Anna Zvaigzne, Aldis Butlers, Inga Ciproviča

**Affiliations:** 1Faculty of Agriculture and Food Technology, Latvia University of Life Sciences and Technologies, Rīgas 22, LV-3001 Jelgava, Latvia; 2Institute of Horticulture, Graudu 1, Ceriņi, LV-3701 Dobele, Latvia; 3Latvian State Forest Research Institute “Silava”, Rīgas 111, LV-2169 Salaspils, Latvia

**Keywords:** de-oiling, pome fruit, valorization

## Abstract

Fruit seeds are often an underutilized side-stream of fruit processing. The most common approach to seed valorization is oil extraction due to the relative simplicity of the process. The partially or fully defatted seed meal is rarely further processed, even though seeds generally contain more protein and fiber than oil. The present study used single-screw extrusion (oil press), supercritical CO_2_ extraction, and a combination of the two, to defat Japanese quince (*Chaenomeles japonica*) seeds, and evaluated the defatted meals as sources of functional protein. Defatting with oil press and CO_2_ extraction proved similarly effective (reduced seed flour fat content from 11.75% to 6.40% and 5.32%, respectively); combining the two methods reduced fat content to 0.90%. The yield was minimally affected, but protein extract purity was defined by defatting efficiency (65.05% protein from non-defatted versus 82.29% protein from a combination-defatted meal). Defatting did not significantly affect amino acid composition but had a significant effect on every tested functional property (solubility, water, and oil binding capacity, apparent viscosity, foaming capacity, and emulsifying activity index). Of the tested defatting methods, supercritical CO_2_ extraction and the combination provided the best results from most aspects.

## 1. Introduction

Japanese quince (*Chaenomeles japonica* L.) is a flowering bush in the Rosaceae family with tart yellow apple-like fruits (pomes) primarily grown in Eastern and North-Eastern Europe. It is one of several quince species used as fruit crops. These include true quinces (*Cydonia oblonga*), Chinese quince (*Pseudocydonia sinensis*), and other flowering quinces, like *Chaenomeles speciosa*, which is also often referred to as Chinese quince or Japanese quince. Like apples and crab apples, different quinces are not interchangeable in processing operations, nor are their by-products, but the species is not specified on products, and separate production data are not available. The largest producers of quinces are Turkey (192,237 tons), China (estimated 111,576 tons), Uzbekistan (95,654 tons), Iran (estimated 90,477 tons), and Morocco (43,523 tons), according to the Food and Agriculture Organization of the United Nations 2023 data. Most of the globally produced quince is *Cydonia oblonga*, but Japanese quince is more popular in colder climates, like Poland, Lithuania, Latvia, and Estonia, where the true quince may not fully ripen during summer. Unlike most fruits, almost all quince is processed because the fruit is generally too tart for fresh consumption. Typical processing products include candied fruit, juice, syrup, and wine. Although production is generally well stream-lined, and almost all of the fruit is used, the seeds are not widely processed. A distinguishing characteristic between certain quinces is the presence of a polysaccharide gum surrounding the seeds. While a generous amount of gum (mucilage) is contained in true quince cores [1], Japanese quince contains very little, making them more suitable for protein extraction without prior washing.

The most common approach to fruit seed processing is oil production, and Rosaceae fruit seed oils are popular in cosmetic formulations, popular examples include almond, peach kernel, and cherry oil. Like other fruits, JQ seeds are relatively rich in oil (10.5–13.2% dw, depending on variety), which predominantly contains linoleic (47.21–51.93% of total fatty acids) and oleic (36.21–40.97% of total fatty acids) acids. They are also rich in α-tocopherol (up to 119.74 mg 100 g^−1^ oil), phytosterols (up to 9.51 g 100 g^−1^ oil), and squalene (1.04–1.50 g 100 g^−1^) [2,3]. However, oil extraction does not fully use up the biomass regardless of the extraction method. The next step in defatted seed processing is protein extraction [4,5], either for dietary or technological production. Widely processed fruits with especially protein-rich seeds include papaya (32% protein of dry weight, dw), pumpkin (48%, dw) [6], oranges (34%, dw), and peaches (34%, dw) [7]. Japanese quince seeds contain 24.6–33.2% protein, depending on the variety [8].

In the case of some crops, protein extraction can be difficult due to low protein content or the presence of antinutrients. While the use of rapeseed protein in food and cosmetics is limited by the presence of antinutrients and protein allergenicity [9], in fruits of the Rosaceae family, there is the presence of amygdalin, a cyanogenic glucoside. However, amygdalin content is severely reduced in the protein extract, for example, apricot-defatted cake protein extract is less than 0.005 mg g^−1^ in [10], while sweet apricot fresh kernel amygdalin content is 0.16 mg g^−1^ [11] to 14.37 mg g^−1^ [12].

The most often used procedure for protein production from seeds is extraction under alkaline conditions to increase protein solubility, followed by lowering the pH to the isoelectric point of most of the proteins in the slurry. The oil extraction method affects oil yield, composition (lipid classes, bioactive compound content, and other minor constituents such as water, protein, and waxes), and quality parameters (oxidative stability and polycyclic aromatic hydrocarbon concentration) [13,14]. As a result of thermal and chemical processes, compounds in the defatted flour are affected as well, such as changed macromolecular compound structure. This can result in different extractability, purity, and functional properties of the protein [15].

The present study aims to evaluate the differential effect of common oils extraction methods (extrusion, supercritical CO_2_ extraction, and solvent extraction using *n*-hexane) on protein extracts from JQ seeds to discern which oil extraction methods provide suitable material for further by-product valorization. This includes the technological and functional properties of the protein extractable from the press cake or defatted flours as well as the interchangeability of defatted flours from different oil extraction operations for the extraction of protein.

## 2. Materials and Methods

### 2.1. Plant Material and Reagents

Mixed-variety Japanese quince (JQ) seeds were collected from a semi-industrial processing operation in Dobele, Latvia, washed, and stored frozen at −18 °C until further processing. Prior to drying, the seeds were thawed in warm potable water, thoroughly rinsed, removing any fruit remains, and then dried in a convection dryer at 45 °C for 24 h. Sodium hydroxide, hydrochloric acid, and *n*-hexane were purchased from Sigma-Aldrich (Steinheim, Germany).

### 2.2. Defatting

For screw-pressing (CP), a pilot-scale Farmet Uno (Farmet, Česká Skalice, Czech Republic) single-screw oil press was used, previously heating the extraction camera to 55 °C. The press cake was collected and stored in a freezer at −18 °C. Seeds and press cake were milled before supercritical fluid extraction (SFE), and pure CO_2_ was used as the extraction medium in a semi-industrial extraction unit SFE 1000 (Faneks, Riga, Latvia). Extraction conditions were as follows: extractor 17,000–17,300 kPa at 50 °C, first separator 10,800–11,000 kPa at 50 °C, second separator 6000–6500 kPa at 40 °C. The same conditions were followed for seeds and CP press cake.

After defatting, several JQ seed flours were obtained: non-defatted seed flour (NDF), milled press cake flour (CP), supercritical CO_2_-defatted flour (SFE), and CP flour subsequently defatted using supercritical CO_2_ (CP-SFE). The seeds and press cakes were milled using a lab-scale grain mill, the flour was sieved through a 2 mm strainer to remove whole and partially milled seeds.

### 2.3. Protein Extraction

Non-defatted and defatted flours were dispersed in distilled water 1:10 (*w*/*v*), and pH was adjusted to 10 using 1 M NaOH (initial pH of slurries was approximately 5.4–5.6). Constantly stirring on a magnetic stirrer, the slurry was extracted at ambient temperature for 2 h. Then, the liquid portion was decanted and centrifuged at 7000 rpm for 10 min at ambient temperature, and the liquid fraction with dissolved protein was collected. The extraction was repeated on the remaining slurry using the same solid:water ratio, based on the removed volume of liquid, for 1 h at pH 10, while constantly stirring. The liquid was decanted from the slurry, centrifuged the same as the first extraction, and the liquid fraction was collected. The pH of the collected supernatant was adjusted to 4.2 using 1 M HCl, centrifuged at 8000 for 8 min, and the supernatant removed. The protein cake was washed two times with distilled water, centrifuged under the same conditions, collected, frozen, and lyophilized for 24 h. The *Yield* (%) was calculated according to formula:(1)$$Yield,\;\%=\frac{m_{flour}}{m_{protein\;cake}}\times 100\%,$$
where *m_flour_*, mass of flour (seed or defatted meal) used for extraction, g;

*m_protein cake_*, mass of protein cake after lyophilization, g.

Protein recovery is the proportion of the recovered protein from the protein cake relative to the protein in the flour used for extraction. Protein recovery was calculated according to the formula:(2)$$Recovery,\%=\frac{m_{protein\;cake}\times c_{protein\;in\;cake}}{m_{flour}\times c_{protein\;in\;flour}}\times 100\%,$$
where *m_protein cake_*, mass of protein cake after lyophilization, g;

*c_protein in cake_*, protein content in cake used for extraction, %;

*m_flour_*, mass of seed flour (defatted or non-defatted) used for protein extraction, g;

*c_protein in flour_*, protein content in seed flour (defatted or non-defatted), %.

### 2.4. Proximate Analysis

Moisture content in the seeds and flours was determined using a KERN MLS 50-3 moisture analyzer (KERN, Landkern, Germany).

#### 2.4.1. Fat Content Determination

For fat content determination, pure *n*-hexane was used as the solvent. Ground seeds or defatted flour (5 g) were weighed in a tube, and 25 mL *n*-hexane was added. The tubes were vortexed for 1 min, then placed in a Sonorex RK 510 H ultrasonic bath (Bandelin electronic, Berlin, Germany) at 35 °C for 5 min. After vortexing for another 15 s, the tubes were centrifuged at 9000 rpm, at ambient temperature for 5 min. The supernatant was gathered in a round-bottom flask. The same extraction protocol was repeated another two times for the residue. The solvent was evaporated from the combined supernatant using a Laborota 4000 rotary vacuum evaporator (Heidolph, Schwabach, Germany) at 40 °C 145 kPa until constant flask weight. Ground seed residue was stored refrigerated until all solvents had evaporated.

#### 2.4.2. Protein Content Determination

For protein content determination in seed flour, defatted and non-defatted, and protein extracts, the nitrogen combustion (Dumas method), and the almond nitrogen–protein conversion coefficient of 5.18 was used.

#### 2.4.3. Amino Acid Analysis

JQ seed and protein extract amino acid compositions were analyzed according to ISO 13,903:2005 [16]. Powdered seeds and protein extracts were weighed in a glass vial with a screw cap and hydrolyzed in a HCl solution in the presence of phenol. Hydrolysis was performed in a closed vial, boiling the mixture for 24 h.

Because tryptophan is degraded into indole during hydrolysis in concentrated HCl, its content could not be determined with the used method, and a separate alkaline hydrolysis process is needed. According to the existing literature, it constitutes approximately 1% of JQ seeds [8,17]. Total amino acid content was expressed as the sum of analyzed amino acids.

### 2.5. Protein Solubility

A 1:100 dispersion (*w*/*v*) of protein extracts was prepared, and the pH was adjusted to 4, 7, and 10 using 1 M HCl and 1 M NaOH solutions. Aliquots of 2 mL were taken, centrifuged at 8000 rpm for 10 min. Afterward, 1 mL of the supernatant was taken and mixed with 2 mL Biuret reagent and briefly vortexed. After 30 min, absorbance of the resulting solution was measured at 540 nm on a Shimadzu 1650 PC spectrophotometer (Shimadzu, Kyoto, Japan). Bovine serum albumin was used as a standard solution in 0 to 1% concentration (m/v). Protein *solubility* was calculated using mean extract protein content from three extraction batches:(3)$$protein\;in\;dispersion,\;\%=\frac{1}{100}\times extract\;protein\;content,$$(4)$$protein\;solubility,\;\%=\frac{protein\;in\;supernatant}{protein\;in\;dispersion}\times 100,$$

### 2.6. Water-Holding Capacity

Protein extract powder water-holding capacity was determined using a method based on the one described by [18]. In triplicate, protein extract powder (1 g) was weighed in a centrifuge tube, deionized water was added (20 mL), and the pH was adjusted to 7 using 1 M HCl or NaOH solutions. The tube was vortexed for 2 min to thoroughly mix the dispersion and left still at ambient temperature for 1 h. Afterward, the contents were centrifuged at 6000 rpm for 10 min, and the liquid was decanted. The procedure was carried out on protein extract at unchanged pH as well as protein extract with pH adjusted to 5, 7, and 9. Water-holding capacity (*WHC*) was calculated according to the following formula:(5)$$WHC,\;g\;g^{-1}=\frac{w_{2}-w_{1}}{w_{0}}$$
where *W*_0_, sample mass, g;

*W*_1_, tube + sample mass, g;

*W*_2_, tube + protein precipitate after centrifugation, g.

### 2.7. Oil Adsorption Capacity

Protein extract powder oil adsorption capacity (OAC) was determined similarly to [18]. Test tubes were weighed, and protein extract powder (1 g) was added in triplicate. Sunflower oil (15 mL) was added and the tube contents were thoroughly mixed with a vortex mixer. The mixture was let undisturbed for 1 h and subsequently centrifuged at 6000 rpm for 10 min. The supernatant was decanted, and the solids were weighed. Because some oil remains on the walls of the test tube, the procedure was performed on three test tubes without protein extract. The oil adsorption capacity was calculated accordingly:(6)$$OAC,g\;g^{-1}=\frac{m_{2}-m_{1}-m_{3}}{m_{0}},$$
where *OAC*, oil adsorption capacity, %;

*m*_0_, mass of protein extract, g;

*m*_1_, mass of test tube before adding oil, g;

*m*_2_, mass of test tube with protein extract and oil, g;

*m*_3_, mass of test tube with oil, g.

### 2.8. Apparent Viscosity

Protein dispersions were prepared in 5% concentration (*w*/*v*), adjusting the pH to 7, using 1 M NaOH, and refrigerated overnight at 4 ± 2 °C, allowing the protein to fully hydrate. Before analysis in a viscosimeter, the dispersions were allowed to warm up to room temperature. Measurements were performed in a Brookfield rotation viscosimeter DV-III Ultra (AMETEK, Berwyn, PA, USA) in triplicate.

### 2.9. Foaming Capacity and Foam Stability

Protein in water dispersions (1:100, *w*/*v*) was prepared, allowed to hydrate at ambient temperature for 1 h under constant agitation on a magnetic mixer, and pH was adjusted to 4, 7, and 10 using 1 M NaOH or HCl solutions. Duplicate aliquots (10 mL) from two extraction batches, each, were transferred into centrifuge tubes for a total of four replicates. The tubes were whipped at 14,000 rpm for 2 min using an immersion high-speed homogenizer IKA T25 (IKA-Werke GmbH & Co, Staufen, Germany), simultaneously cooling the centrifuge tube in cool water. The homogenizer head was removed and the liquid and foam level (volume indicator) were noted immediately after whipping, and 30 min after whipping. Foaming capacity (*FC*) and foam stability (*FS*) were calculated according to the following formulas:(7)$$FC,\;\%=\frac{V_{0}}{V}\times 100,$$(8)$$FS,\;\%=\frac{V_{30}}{V_{0}}\times 100,$$
where *V*, dispersion volume before foaming, mL;

*V*_0_, initial foam volume, mL;

*V*_30_, foam volume after 30 min, mL.

### 2.10. Emulsifying Activity Index and (EAI) Emulsion Stability Index (ESI)

A method initially described by [19] and slightly modified by [18] was followed with reduced volumes for emulsion preparation. Protein extracts were dispersed in distilled water (1 g:100 mL) and allowed to hydrate for 1 h under constant agitation on a magnetic mixer at ambient temperature. The pH of the dispersion was gradually adjusted to 4, 7, and 10 using 1 M NaOH and HCl solutions, taking duplicate aliquots (10 mL) at each pH value into centrifuge tubes. The procedure was performed using two separate extraction batches for a total of four replications at each pH value. Commercial refined sunflower oil (3.33 mL) was added to each centrifuge tube, and the mixture was homogenized using a high-speed immersion homogenizer (IKA T25) for 2 min at 14,000 rpm, simultaneously cooling the centrifuge tube in cool water. Immediately after homogenization, 50 μL were taken from the bottom of the centrifuge tube and transferred into a tube containing 5 mL 0.1% sodium dodecyl sulfate (SDS) and mixed on a vortex mixer. Another 50 μL aliquot was taken from the bottom of the undisturbed emulsified centrifuge tube, 30 min after emulsification. The absorbance of the emulsion aliquots in SDS solution was measured at 500 nm in a Shimadzu 1650 PC spectrophotometer (Shimadzu, Kyoto, Japan). The emulsifying activity index (*EAI*) and the emulsion stability index (*ESI*) were calculated as follows:(9)$$EAI\;(m^{2}\;g^{-1})=\frac{2\times 2.303\times A_{0}\times dilution\;factor}{C\times \varphi \times 1000},$$(10)$$ESI\;(\%)=\frac{{EAI}_{30\;\min}}{{EAI}_{0\;\min}\;}\times 100$$
where *A*_0_, absorbance of emulsion in SDS solution taken immediately after emulsification;

*dilution factor* = 100;

*C*, the concentration of the sample = 10 mg mL^−1^;

*φ*, oil volumetric fraction, 0.25;

*EAI_3_*_0_, EAI calculated using absorbances of emulsion aliquots in SDS solution immediately and 30 min after emulsification.

### 2.11. FTIR Spectral Analysis

Non-defatted seed flour and protein extract powders were scanned in a Bruker Invenios-S mid-infrared diffuse reflectance Fourier-transform (MIR-DRIFT) spectrometer (Bruker, Billerica, MA, USA) in three powdered aliquot replicates without any additional processing. Principal component analysis (PCA) was performed to analyze the spectral data using open-source R library factoextra with scaled absorbance readings to identify spectral regions with the highest variability through principal component loadings.

### 2.12. Statistical Analysis and Visualization

For variables independent of pH, one-way analysis of variance (ANOVA) was used to determine statistically significant differences between defatting methods with pairwise *t*-test post hoc and Bonferroni *p*-value adjustment to determine which defatting methods produced different results from each other. Differences were considered statistically significant when the *p*-value was equal to or lower than 0.05. If variables were observed across several pH values, multivariate analysis of variance (MANOVA) was used to determine which factor (pH or defatting method) had a strong effect on the protein property with Tukey post hoc test and Bonferroni *p*-value adjustment. Relationships between quantitative variables were examined using linear regression analysis. Statistical analysis was performed using R packages stats and factoextra in RStudio build 764 “Kousa Dogwood” Release (cf37a3e5488c937207f992226d255be71f5e3f41, 2024-12-11) (Posit Software, Boston, MA, USA). Data visualizations were produced and formatted using R packages ggplot2, dplyr, ggthemes, and forcats.

## 3. Results

### 3.1. Yield, Recovery, Protein Extract Purity, Composition, and Spectrum

The non-defatted seed flour contained 19.86% protein and 11.75% fat (Table 1). After pressing oil in an oil press, the fat content decreased to 6.40%, while supercritical CO_2_ reduced the fat content to 5.32%. The flour fat content was reduced to 0.90%, using a combination of the two methods. As a result, the protein proportion in the flour increased after defatting, and the widest difference–22.63% protein in SFE-defatted seed flour–is equal to a 17.88% increase in protein content.

The defatting method affected protein yield (*p* = 0.024), purity (*p* = 0.0021), and recovery (*p* = 0.00080) with statistical significance (Appendix A). However, the *t*-test only showed statistically significant differences between SFE-defatted and non-defatted seed flour (*p* = 0.047) and SFE-defatted and oil press-defatted seed flour (*p* = 0.0017) protein yield. While statistical analysis did not show other significant differences, the SFE treatment appears to slightly reduce protein yield—mean SFE-defatted flour is slightly lower than untreated flour (by 1.39%, *p* = 0.047); combination-defatted flour protein yield was lower than oil press-defatted flour (by 0.73%, *p* > 0.05) protein yield. This may be partially explained by some protein denaturation during the SFE treatment as a result of prolonged increased temperature and pressure. However, non-defatted seed flour was similar to the yield from oil press-defatted flour, which was exposed to higher increased temperature and pressure, albeit for a shorter time. Alternatively, it may be caused by the lower content of impurities such as residual lipids in the SFE-treated flour proteins. Existing studies report increased negative ζ-potential in plum kernel proteins treated with SFE at elevated temperatures up to 60 °C, and reduced negative ζ-potential at 70 °C, while turbidity increased at a higher treatment temperatures [20], and egg whites treated with supercritical CO_2_ at 6 to 20 MPa pressure had decreased negative ζ-potential [21]. Low-surface charge (ζ-potential) can be related to low solubility [18], but SFE-treated lupin protein isolate solubility did not differ from untreated protein [22], nor did SFE and hexane-defatted tomato seed protein extracts [23].

The yields are proportional to those produced from Japanese quince wine by-products (pomace and seeds), where a 6.04% protein isolate yield was obtained from pomace containing 11.42% protein following ethanol-assisted supercritical CO_2_ extraction [17].

Similarly to yield, statistically significant differences in protein purity were only detected between protein extracts from non-defatted and combination-defatted (*p* = 0.0019), SFE-defatted (*p* = 0.018) protein extracts, and a marginally significant difference between non-defatted and oil press-defatted seed flour (*p* = 0.079). Any defatting method had a positive effect on protein purity, but differences were not significant between defatting methods (*p* > 0.05). Defatting had a distinct positive and proportional effect on protein purity following a linear trend (R^2^ = 0.78 using mean values). The linear fit is better aligned with the protein content in flour (R^2^ = 0.99 using mean values), but the protein content in flour is determined by defatting efficiency since protein losses during defatting are negligible.

The protein extracts from non-defatted and oil-press defatted seed flour can be categorized as protein concentrates at 65.06 and 76.25% protein, while a protein isolate was technically produced from SFE and combination-defatted seed flour at 83.88 and 82.29%. The purity of the protein isolates is in line with similar material—an 87.56% protein isolate has been produced from the closely related Chinese quince (*Chaenomeles speciosa*) seeds defatted, using petroleum ether in a Soxhlet unit [18]. A team [24] investigating the effect of defatting on oat protein extractability and properties also found that defatting with hexane improves protein isolate purity, improving protein isolate purity by approximately 20% wet basis, although extraction yield and protein recovery were decreased. A dry basis of 11–19% yield and 62–68% recovery was produced from non-defatted oat flour, but these figures were reduced to 6–14% yield and 39–59% recovery after defatting—the increased protein purity did not compensate for the reduced yield [24].

Statistically, differences in protein recovery were more pronounced—all tested defatting treatments resulted in increased protein recovery; the highest recovery was achieved using oil press (38.46%) and combination-defatted (30.01%) seed flour. This is a cumulative result of the slightly increased flour protein content, protein yield, and much-improved protein extract purity. Recovery was not proportional to the residual fat content in the flour (R^2^ = 0.54 using mean values). The recovery values are within previously observed ranges when using similar extraction protocols, including oat [24], peach [25], and hemp seeds [26].

As shown in Table 2, all analyzed proteinogenic amino acids were present in Japanese quince seeds as well as the protein extracts, and the major amino acids from analyzed amino acids were glutamic acid (29.12–31.58%), arginine (10.63–12.35%), aspartic acid (9.89–11.11%), and leucine (6.93–7.56%). The amino acid profile was statistically significantly different between defatting methods, but differences were very minor and within a couple of percent for major amino acids. While tryptophan content was not analyzed in the present study, other authors have observed that the tryptophan proportion is approximately 1% [8,17] of total amino acids in Japanese quince seeds and seed protein.

Individual amino acids and amino acids grouped by side chain did not differ by more than a few percent, which is within the natural variability of seed protein composition [24]. Since free amino acid content in plant seeds is very low, protein molecular fraction solubility would need to be affected first. While no electrophoretic profile could be produced for the present study, the results of other research teams shows that the choice of the defatting method impacts the solubility of the whole protein fraction and less the solubility of individual fractions when comparing defatting soy flour defatted with hexane or supercritical CO_2_ [27]. Differences in amino acid contents have previously been observed between hexane and oil press-defatted seed flour proteins, but electrophoretic (SDS-PAGE) profiles were similar [28]. Using different protein extraction methods on the same plant material, results are similar. The major amino acid proportion differed by only a few percent in pressurized liquid extraction, microwave-assisted extraction, and heat-stirred extraction of peach kernel protein isolates [25]. A significant effect on hemp seed protein molecular fraction solubility has been observed at different extraction pH [29].

Essential amino acids constituted 26.49 (combination-defatted)–29.09% (non-defatted seed flour) of analyzed amino acids. Essential and partially essential amino acid proportion in comparison to World Health Organization recommendations circa 2007 [30] is provided in Figure 1. The essential amino acid proportion is similar to other plant protein sources and fruit proteins. Oat protein isolate contains 35–38% of essential amino acids [24], apricot, peach, and mango protein isolates contain 27–36% of essential amino acids [31]. As a potential dietary protein, the protein quality of Japanese quince is limited by low lysine and methionine proportion, which is typical for seeds and legumes. On the other hand, phenylalanine + tyrosine is relatively abundant. However, the use of Japanese quince seed proteins as a major source of dietary protein is unlikely due to the expected high cost.

The mean values of seed flour and protein extracts FTIR spectrums are provided in Figure 2 along with the average absorbance, principal component 1 (PC1), and principal component 2 (PC2) loading at each wavenumber. The establishment of a baseline absorption in the amide I and amide II was unsuccessful, possibly due to the drying of proteins close to their isoelectric point at which protein secondary structures are generally not formed. Secondary protein structures are formed by the association of charged amino acid side-chains on a protein molecule. The isoelectric point of a protein is defined by the lowest surface charge and amino acid side chains are generally not charged. Alternatively, the spectral peaks associated with specific secondary protein structures overlap significantly. This makes establishing a reflectance baseline and analysis difficult without specialized analytical software.

Seed flour, non-defatted flour protein, and oil press-defatted flour protein had higher absorbance in the CH stretch range (2800–3000 cm^−1^), indicating the presence of residual lipids in the protein extracts. They were also differentiated by an absorbance peak at around 1742 ± 2 cm^−1^, which is outside the amide absorption range. SFE and combination-defatted flour protein extract had peaks at around 1700 cm^−1^, the upper boundary of the amide I range. According to PCA, the proportion of explained variance for PC1 and PC2 is 80.31% and 14.31%, respectively, for a cumulative explained proportion of variance of 94.62% with two principal components. However, loadings across the wavenumber spectrum are close to 0, indicating little difference between chemical bonds in the samples. Loadings were close to 0, indicating a small effect on variability and little variance across the spectrum. A relatively high loading for PC1 was identified at around 1700 cm^−1^, the higher limit of the amide I range. The boundary between the amide I and amide II range and the lower amide II range boundary showed slight negative loading peaks. While PC2 explains far less of the variation, several positive loading peaks are identified: several overlapping peaks in the amide A, amide I, amide II, and amide III range. While the amide I, II, and III regions are associated with the secondary structure of proteins, absorbance in the amide A and B regions is associated with amino acid side-chains.

Seed and protein extract PC1 and PC2 score plots are provided in Figure 3. There is no apparent relation to the protein content or expectable residual fat in the sample, or degree of processing, as the combination-defatted protein extract spectrum appears more similar to the native seed and non-defatted flour protein extract spectrum than oil press or SFE-defatted flour protein extract. However, the whole spectrum was used to calculate PC1 and PC2 loadings, and PC scores do not accurately reflect the similarities and differences between the spectral readings. Compared to the mean absorbance, both SFE-treated flour protein extracts are distinctly different from seed flour and protein from flour that was not treated with SFE (Appendix A). The parallel clusters of non-defatted flour protein and seed flour (lower PC1 and PC2 scores) and oil press and SFE-defatted flour protein (higher PC1 and PC2 scores) imply some connection to residual fat content; however, this is unlikely due to the placement of combination-defatted flour protein between the two, since combination-defatted flour had the lowest fat content, the difference between the fat content in the seeds and non-defatted flour protein extract, and the low absorbance in the CH stretch range of the combination-defatted flour protein.

### 3.2. Functional Properties

#### Solubility

Protein solubility differed among the defatting methods (*p* > 0.05), and the interaction between the defatting method and pH was significant (*p* < 0.001). Solubility was similar between protein extracts at pH 4 (*p* > 0.05), and only non-defatted and combination-defatted flour proteins were statistically different (*p* < 0.05) at pH 10. Oil press and combination-defatted flour protein solubility were higher at pH 7. Solubility at different pH values is provided in Table 3. Observed protein solubility increased at higher pH following a moderately exponential trend.

The isoelectric point of closely related Chinese quince seed proteins is around 4.7, and higher solubility of protein at higher pH is expected and well documented [18]; however, the provided values are affected by the method used for the quantification of dissolved protein. The copper reagent used in the Biuret assay reacts with the side-chains of aromatic amino acids (tyrosine, phenylalanine, and tryptophan) as well as the aromatic groups in phenolic compounds, including tannins, which can co-extract and form complexes with proteins, potentially leading to values above 100% dissolved protein when both protein and phenolic molecules are present in the solute, as is in the present study.

At pH 10, the different solubility may be explained by the removal of phenolic compounds during SFE treatment or reduced protein solubility as a result of the SFE treatment. Existing research has shown that only quinic acid is removed during SFE of Japanese quince pomace with or without added ethanol [17]. SFE-treated lupin flour protein had a lower polyphenol content than untreated flour protein, but solubility was unaffected across a range of pH [22]. Although no detailed data exists for quince seeds, tannins are generally concentrated in the seed coat, or hull—dehulling may help reduce tannin content in the raw material for protein extraction; however, existing findings indicate a limited effect on protein secondary structures, ζ-potential or solubility [26].

Unrealistic values are not caused by drastically different amino acid side-chain composition. BSA contains about 11.52% aromatic side-chain amino acids [32], which is higher than the 7.67% in Japanese quince seeds—a BSA solution of equal protein content would produce a slightly stronger color. Alternatively, the absorbance reading may have been increased by cloudiness imperceptible to the naked eye as a result of residual fat and dissolved protein forming an emulsion.

Water-holding capacity (WHC), oil adsorption capacity (OAC), and viscosity.

Water-holding capacity and oil adsorption capacity of the protein extracts are provided in Table 4 and depicted in Appendix A. Water-holding capacity was statistically significantly different between differently defatted flour proteins (*p* < 0.001) and native versus neutralized pH (*p* < 0.001) without statistically significant interaction between the two factors (*p* = 1). The different WHC values are proportional to protein extract purity at native (R^2^ = 0.82) and neutralized (R^2^ = 0.83) pH, but cannot be explained entirely by protein purity, since carbohydrate impurities would adsorb water as well, and thermal treatment can affect the WHC of both through partial denaturation (protein) or gelatinization (carbohydrates and some proteins at sufficiently high temperatures). WHC of the protein at native pH was higher than when neutralized, but this is likely due to proteins dissolving.

Oil adsorption capacity was near-significantly higher in the combination-defatted flour protein extract (*p* = 0.062–0.083), while OACs of the other protein extracts were similar (*p* > 0.05). Combination-defatted flour protein had the highest OAC, while those of the non-defatted, oil press, and SFE-defatted flour protein extracts had similar OAC. While WHC was somewhat proportional to protein purity, OAC was not and was not explained by a cumulative effect of low residual fat and high protein content.

The defatting method had a significant effect on apparent viscosity (*p* < 0.001), although SFE and combination-defatted flour protein extract’s apparent viscosities were similar (*p* > 0.05). Unlike WHC and OAC, the highest apparent viscosity was observed in oil press-defatted flour protein extract. This may have been caused by residual lipids in the protein, since higher fat content is related to higher viscosity, but does not affect all protein dispersions equally. For example, soy and casein high-fat colloidal system viscosity was not significantly different from a low-fat control [33]. However, rapeseed protein-stabilized emulsions made with milk fat or rapeseed oil displayed a fat content-dependent (10–50%) apparent viscosity increase [34]. Like the rapeseed protein emulsions, oil press-defatted protein had higher absorption in the CH stretch region of the FTIR spectrum than SFE or combination-defatted flour protein. However, the cumulative absorbance in the range was lower than that of the non-defatted flour protein, which had much lower viscosity. The higher apparent viscosity is therefore not explained by residual fat alone.

### 3.3. Foaming and Emulsifying Properties

#### Foaming Capacity (FC) and Foam Stability (FS)

The foaming capacity of protein extracts differed significantly between defatting methods and pH values (*p* < 0.001), but the interaction between pH and defatting method was insignificant (*p* > 0.05). A statistically significant effect was not observed between pH 4 and 7 (*p* > 0.05), but was pronounced with pH increased to 10 (*p* < 0.001 between 10 and 4, *p* = 0.0027 between 10 and 7). Overall, combination-defatted flour protein extract differed from oil press defatting and non-defatted protein extract (*p* = 0.0062 and <0.001, respectively). As shown in Table 5 and Appendix A, foaming capacity measurements had a very high standard deviation, mostly related to the test batch, especially combination-defatted flour protein. Non-defatted seed flour protein did not produce foam at pH 4 and produced little foam at pH 7. Oil press and SFE-defatted flour protein produced similar results, and SFE was only higher at pH 10. However, foamability in alkaline conditions is essentially only relevant in cosmetic products, where a variety of more effective and cheaper foaming agents are available and acceptable to the consumer. Likely due to the presence of residual fats in the protein extracts and the anti-foaming agent properties of polar lipids, foaming capacity was low in samples other than the combination-defatted flour protein.

Statistically, foam stability was affected by pH (*p* = 0.018), the defatting method (*p* = 0.030), and the interaction between the two factors was significant (*p* = 0.0038). However, large standard deviations were observed, especially at pH 4 and 7, rendering minor differences.

The foaming properties of the protein are pH-dependent and determined by the surface charge of the protein. Moreover, different protein fractions are dissolved depending on the pH, as are the secondary structures of the protein. Higher foaming capacity has been observed in almond protein dispersions when smaller molecular fractions are dissolved and able to incorporate in the liquid–air interface [35]. High surface hydrophobicity and the proportion of hydrophobic amino acids are also beneficial for producing and maintaining foam [18]. Since the amino acid composition and molecular fractions of the protein dispersions are likely similar, residual fat explains the low foaming capacity.

### 3.4. Emulsifying Activity Index (EAI) and Emulsion Stability Index (ESI)

Protein extract EAI was not affected by the defatting method (*p* = 0.746), but pH had a strong effect on EAI (*p* < 0.001), and interaction between the defatting method and pH was strong (*p* < 0.001). Emulsion stability was similarly affected—the defatting method did not have a statistically significant effect (*p* = 0.28), but ESI was different at pH (*p* < 0.001), and there was a significant interaction between the defatting method and pH (*p* = 0.018). EAI and ESI were less influenced by the protein extraction batch than foaming properties were (Table 6).

The pH-dependent trends differed between differently defatted flour protein extracts (Appendix A). EAI did not differ between defatting methods as significantly as foaming capacity did. Foaming capacity increased linearly as pH increased. EAI increased logarithmically in non-defatted and SFE-defatted flour but plateaued at seven for oil press and combination-defatted protein. Similarly, ESI increased in non-defatted and SFE-defatted protein emulsions but was reduced in oil press and combination-defatted protein.

While the data gathered in the present study do not explain reduced SFE and combination-defatted flour protein EAI, it is possible that the additional period of high temperature and pressure imposed on the seed flour in the extrusion chamber during oil pressing has affected protein molecules either by partial denaturation or cross-linking with other molecules. Considering that the difference between the EAI of the differently defatted flour proteins is not proportional to protein purity or residual fat in the flour, partial protein denaturation is a more likely culprit.

## 4. Discussion

As a novel nutritional protein candidate, Japanese quince seed protein’s nutritional value is similar to most nuts and seeds. Glutamic acid, arginine, aspartic acid, and leucine have the highest proportion of total amino acids, the essential amino acid proportion is within 30–45%, and the limiting essential amino acids are methionine and lysine, similar to apricot, peach, and mango kernels [31], soy products and seitan, except lysine is a limiting amino acid in seitan, valine in soy milk, and sulfur-containing amino acids (methionine and/or cysteine) in tofu and peas [36].

Protein yield and recovery were similar to protein yield produced from other plant seeds, as is the amino acid composition—the protein contains all essential amino acids, but requires complementary proteins rich in lysine and methionine. The economic and logistical aspects of extracting protein from fruit seeds and processing side streams have not been discussed, but they present a complex hurdle in addition to legislative issues. Higher protein yield has previously been achieved by partially hydrolyzing fruit seed meals using proteolytic enzymes, which reduces protein molecule size and increases their solubility [6]. However, the products were not directly compared to an extraction protocol without hydrolysis in terms of nutrition or functionality. The partial hydrolysis of almond proteins with alcalase can significantly reduce molecular weight fractions across 1–9% degree of hydrolysis. Amandin, the main almond storage protein with molecular weight 62–66 kDa, as well as bands at around 40 and 20 kDa are reduced to polypeptides with <20 kDa molecular weights [37]. Alcalase-hydrolyzed almond protein isolates (peptides) had higher surface hydrophobicity, solubility, and foam expansion, but gelation properties were reduced [38]. Besides increased yield and recoverability, protein hydrolysis modifies the biochemical properties of plant proteins. Almond protein possesses lipase inhibitory properties, which are decreased after hydrolysis, while α-glucosidase inhibitory activity is only exhibited by hydrolyzed protein. Antimicrobial effects were tested, but no inhibition was observed [39]. The predominant Japanese quince molecular fractions are at 54.1–57.7 kDa and 70–77.6 kDa with a much weaker band sat 35 kDa. The larger protein fractions are aggregates of around 50, 35, and 20 kDa large proteins, which are evident under reduced conditions on the SDS-PAGE lanes [8]. Chinese quince (*Chaenomeles speciosa*) seed proteins, which are composed of about 60 kDa, above 50 kDa, and about 20 kDa molecular fractions, have been successfully hydrolyzed using papain [40].

Practical upper limits of residual fat in the flour used in extraction or in the protein extract cannot be surmised from the reported results, but the benefits of additional seed meal defatting are clear. The experimental design did not control potential residual lipids in the protein extract, which may have a significant effect on functional properties. It also provides limited insight into the effect of polyphenolic compounds on protein properties. The complexation of polyphenols and proteins has been of growing interest as a way to improve polyphenol and protein functionality [41], but their effect depends on the type of interaction and phenolic compound type. Polyphenol molecules can interact with protein molecules covalently (mainly initiated by oxidation and the formation of quinones) or non-covalently (hydrophobic or electrostatic interaction, or hydrogen bonds without modification of polyphenol or protein molecules), and are capable of protein cross-linking by simultaneously interacting with several protein molecules. Their interaction can be affected by the pH, temperature, and polyphenol–protein ratio [42]. The effect depends on the type of polyphenol. For example, gluten strength is decreased by low molecular weight polyphenols because their antioxidant properties reduce disulfide cross-linking and secondary structure formation, while tannins, which have higher molecular weight, increase gluten molecule cross-linking [43]. Soy protein isolate cross-linking with (-)-epigallocatechin gallate and chitosan increased the size of molecular fractions, altered protein secondary structure proportions, and increased emulsion stability [44]. Cross-linking soy protein isolates with tannic acid improved emulsification activity and emulsion oxidative stability [45]. However, the effect of co-extracted native seed proteins on Japanese quince and other fruit seed proteins has not been widely studied in detail.

Moreover, most protein extraction schemes include a protein neutralization step, which may have a differential effect on protein extract composition and functionality. All are subjects and opportunities for future research.

## 5. Conclusions

Japanese quince seed protein extraction is a viable next step in seed valorization if seed oil is extracted. Oil extraction—defatting—is crucial to producing a high-quality, high-purity protein with desirable properties. Although the defatting method had a minor effect on protein yield, all defatting methods had a positive impact on protein purity and recovery. Supercritical CO_2_ extraction and oil press pressing followed by an additional defatting step produced favorable results in wide areas. The seed protein quality and amino acid composition is similar to other seed proteins, essential amino acids made up 26.49–28.53% of analyzed amino acids in the protein extracts, the limiting amino acids are lysine and methionine, while there is a relative abundance of phenylalanine + tyrosine. The defatting method had a significant impact on solubility, water-holding capacity, oil adsorption capacity, viscosity, foaming capacity, and emulsifying activity index, with SFE and combination-defatted flour protein performing the best in most respects.

## Figures and Tables

**Figure 1 foods-14-00234-f001:**
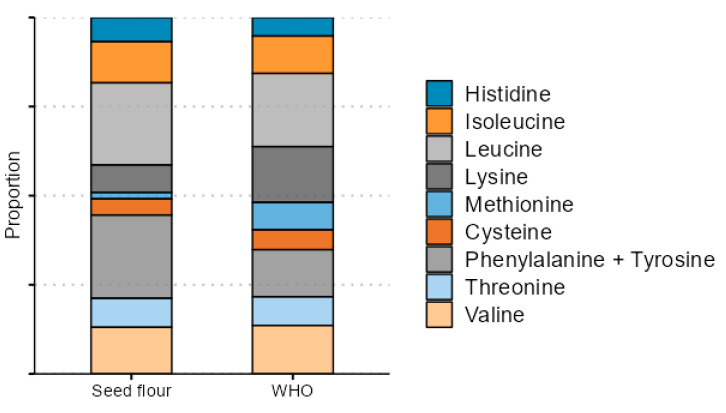
Essential amino acid proportion in Japanese quince seeds (% of analyzed amino acids) and proportional intake recommendations (mg per g protein) according to the WHO [30]. Tryptophan is not included, as it was not analyzed in the study, other essential amino acids are provided in the same top-down order in the legend and plot.

**Figure 2 foods-14-00234-f002:**
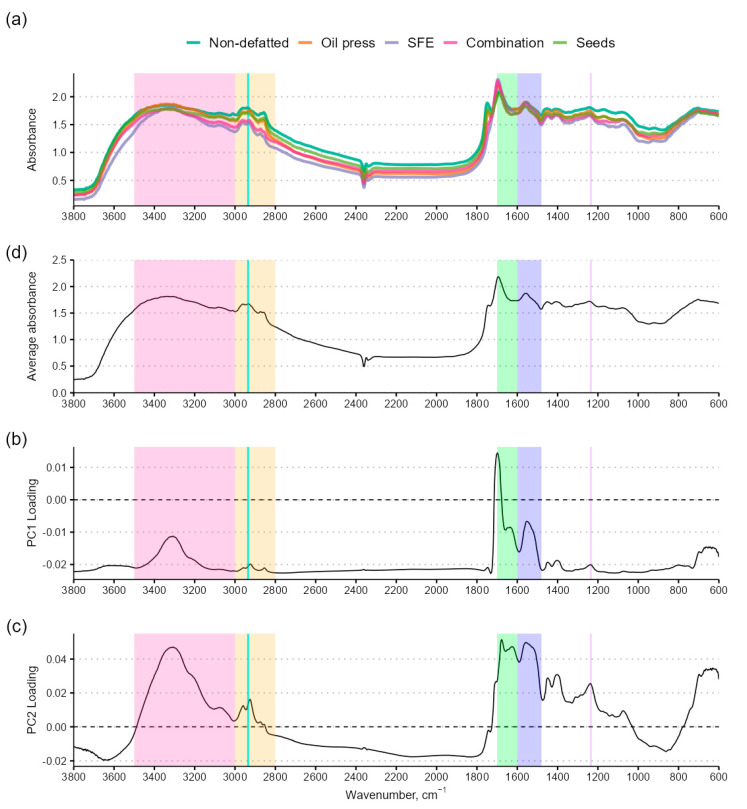
Diffuse reflectance Fourier transform infrared spectroscopy spectra and principal component loading across wavenumber values. (**a**) Mean (*n* = 3) seed flour or protein extract powder absorbance (color legend provided above plot); (**b**) average absorbance across all powdered samples; (**c**) PC1 loadings across wave numbers; (**d**) PC2 loadings across wave numbers. For easier comparison, relevant ranges in the FTIR spectrum are provided as colored bands: amide I (green, 1600–1700 cm^−1^), amide II (blue, 1480–1600 cm^−1^), amide III (purple, 1230–1240 cm^−1^), C-H stretch (orange, 2800–3000 cm^−1^), amide A (pink, 2000–3500 cm^−1^), and amide B (cyan, 2930–2940 cm^−1^).

**Figure 3 foods-14-00234-f003:**
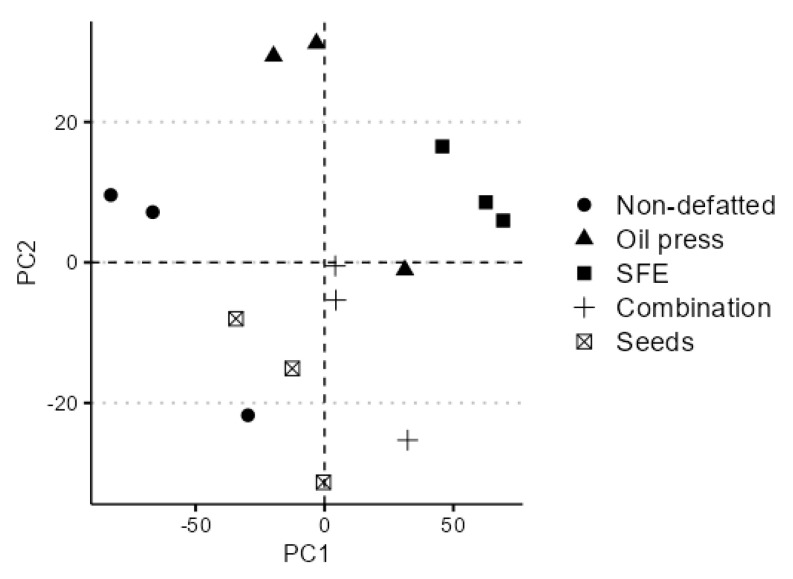
PCA plot of PC1 and PC2 score based on FTIR spectrum. Data are presented as individual readings by their PC 1 and PC 2 score.

**Table 1 foods-14-00234-t001:** Seed flour fat and protein content, extraction protein yield, protein content in protein extracts, and protein recovery from seed flours.

Defatting Method	Fat, %	Protein in Seed Flour, %	Yield, %	Protein in Dried Extract, %	Protein Recovery, %, from Seed Flours
Non-defatted	11.75 ± 0.13	19.86 ± 0.08	11.60 ± 0.43	65.05 ± 2.29	33.36 ± 1.25
Oil press	6.40 ± 0.06	21.18 ± 0.22	11.81 ± 0.63	76.25 ± 1.68	38.46 ± 2.04
SFE	5.32 ± 0.15	22.63 ± 0.53	10.21 ± 0.72	83.88 ± 3.99	35.30 ± 2.48
Combination	0.90 ± 0.16	22.31 ± 0.53	11.08 ± 0.2	82.29 ± 2.15	38.01 ± 0.69

Data are presented as means of two (protein in seed flour) or three (fat content, yield, protein in dried protein extract, recovery) ± standard deviation.

**Table 2 foods-14-00234-t002:** Relative amino acid composition and side-chain proportions in Japanese quince seeds and protein extracts, % of total analyzed amino acids.

Amino Acid	Seeds	Protein Extracts			
		Non-Defatted	Oil Press	SFE	Combination
Essential					
Histidine (His)	2.24 ± 0.37	2.09 ± 0.06	2.07 ± 0.07	2.06 ± 0.09	1.89 ± 0.00
Lysine (Lys)	2.53 ± 0.33	1.79 ± 0.14	1.67 ± 0.09	1.89 ± 0.10	1.61 ± 0.03
Valine (Val)	4.29 ± 0.19	4.22 ± 0.08	4.07 ± 0.16	4.17 ± 0.12	3.94 ± 0.14
Methionine (Met)	0.59 ± 0.05	0.50 ± 0.07	0.57 ± 0.03	0.52 ± 0.01	0.82 ± 0.02
Leucine (leu)	7.56 ± 0.20	7.40 ± 0.08	7.29 ± 0.08	7.47 ± 0.05	6.93 ± 0.06
Isoleucine (Ile)	3.78 ± 0.25	4.11 ± 0.03	3.72 ± 0.04	3.86 ± 0.08	3.69 ± 0.01
Phenylalanine (Phe)	5.43 ± 0.04	5.89 ± 0.12	5.69 ± 0.05	5.71 ± 0.15	5.46 ± 0.04
Total essential amino acids	29.09 ± 0.14	28.53 ± 0.68	27.20 ± 0.8	27.93 ± 0.29	26.49 ± 0.66
Non-essential and partially essential					
Cystine (Cys)	1.49 ± 0.07	0.89 ± 0.01	0.86 ± 0.08	0.99 ± 0.04	0.9 ± 0.01
Serine (Ser)	3.78 ± 0.24	3.83 ± 0.04	3.37 ± 0.13	3.92 ± 0.24	3.66 ± 0.06
Aspartic acid (Asp)	10.49 ± 0.34	11.11 ± 0.21	9.89 ± 0.36	10.36 ± 0.09	10.93 ± 0.02
Glycine (Gly)	5.88 ± 0.25	5.02 ± 0.59	5.19 ± 0.12	5.28 ± 0.08	5.52 ± 0.11
Threonine (Thr)	2.66 ± 0.02	2.53 ± 0.1	2.13 ± 0.04	2.25 ± 0.04	2.14 ± 0.02
Glutamic acid (Glu)	29.12 ± 0.1	29.28 ± 1.13	31.58 ± 0.3	29.93 ± 0.42	30 ± 0.58
Alanine (Ala)	3.54 ± 0.04	4.19 ± 0.07	3.91 ± 0.1	3.9 ± 0.19	3.9 ± 0
Proline (Pro)	3.73 ± 0.12	3.46 ± 0.19	3.15 ± 0.12	3.2 ± 0.25	3.32 ± 0.01
Arginine (Arg)	10.63 ± 0.73	10.68 ± 0.39	12.00 ± 0.05	11.39 ± 0.25	12.35 ± 0.36
Tyrosine (Tyr)	2.24 ± 0.05	3.01 ± 0.06	2.85 ± 0.04	3.11 ± 0.07	2.94 ± 0.05
Proportion by side chain					
Hydrophilic (polar)	65.19 ± 0.63	65.21 ± 0.49	66.42 ± 0.32	65.9 ± 0.52	66.43 ± 0.01
Polar neutral	18.03 ± 0.18	18.4 ± 0.02	16.84 ± 0.12	17.88 ± 0.7	17.68 ± 0.33
Basic	15.4 ± 0.31	14.57 ± 0.85	15.73 ± 0.33	15.34 ± 0.44	15.86 ± 0.2
Acidic	39.62 ± 0.92	40.38 ± 0.41	41.47 ± 0.56	40.29 ± 0.33	40.93 ± 0.06
Hydrophobic (non-polar)	34.81 ± 0.63	34.78 ± 0.47	33.58 ± 0.32	34.11 ± 0.53	33.58 ± 0
Aromatic	7.67 ± 0.07	8.90 ± 0.23	8.53 ± 0.01	8.83 ± 0.22	8.4 ± 0.01
Branched	15.64 ± 0.03	15.73 ± 0.29	15.08 ± 0.21	15.5 ± 0.14	14.56 ± 0.12

Data are presented as means ± standard deviation of two readings. Amino acid sums by side chain were grouped as follows: Hydrophilic (Cys, Ser, Asp, Thr, Glu, His, Lys, Arg, Tyr); Polar neutral (Cys, Ser, Thr); Basic (His, Lys, Arg); Acidic (Asp, Glu); Non-polar (Gly, Ala, Pro, Val, Met, Leu, Ile, Phe); Aromatic (Tyr, Phe); Branched (Leu, Ile, Val).

**Table 3 foods-14-00234-t003:** Solubility of protein extracts at different pH, % of theoretically dissolvable protein.

Defatting Method	pH		
	4	7	10
Non-defatted	15.14 ± 4.17	62.49 ± 11.08	164.90 ± 21.09
Oil press	7.90 ± 1.39	96.07 ± 10.28	148.49 ± 3.68
SFE	15.36 ± 1.75	59.58 ± 20.76	131.86 ± 9.58
Combination	6.70 ± 2.38	95.59 ± 23.10	132.12 ± 8.44

The data are presented as means ± standard deviation of two replicates from two extraction batches, each (*n* = 4).

**Table 4 foods-14-00234-t004:** Water-holding capacity (%), oil absorption capacity (%) of Japanese quince seed protein extracts, and apparent viscosity (Pa s) of 5% protein dispersions at neutralized pH.

Defatting Method	WHC		OAC	Viscosity
	Native pH	pH 7	Native pH	pH 7
Non-defatted	150.57 ± 4.86	98.41 ± 4.98	47.92 ± 17.03	2.33 ± 0.05
Oil press	206.39 ± 17.25	157.07 ± 17.69	50.66 ± 39.91	4.26 ± 0.05
SFE	236.08 ± 23.88	186.17 ± 24.76	51.98 ± 25.28	3.47 ± 0.20
Combination	282.81 ± 13.28	231.85 ± 12.9	116.79 ± 5.76	3.28 ± 0.06

Data are presented as means ± standard deviation of three replications (*n* = 3).

**Table 5 foods-14-00234-t005:** Foaming capacity and foam stability of 1% Japanese quince seed protein extract dispersions at different pH, %.

Defatting Method	Foaming Capacity			Foam Stability		
	pH			pH		
	4	7	10	4	7	10
Non-defatted	nf	17.50 ± 12.58	42.50 ± 5.00	nf	64.58 ± 29.17	78.13 ± 21.35
Oil press	22.50 ± 6.45	36.25 ± 9.46	72.50 ± 22.17	65.84 ± 12.29	73.10 ± 25.95	80.94 ± 13.52
SFE	20.00 ± 8.16	38.75 ± 2.5	122.5 ± 42.52	70.83 ± 34.36	46.43 ± 39.42	53.13 ± 10.27
Combination	45.00 ± 12.91	97.50 ± 60.76	190 ± 139.76	57.09 ± 21.45	79.17 ± 17.72	73.42 ± 13.57

Data are presented as means ± standard deviation of two replicates from two extraction batches, each (*n* = 4). Abbreviations: nf, no foam.

**Table 6 foods-14-00234-t006:** Emulsifying activity index (EAI) and emulsion stability index (ESI) of 1% Japanese quince seed protein extract dispersions at different pH.

Defatting Method	EAI, m^2^ g^−1^			ESI, %		
	pH			pH		
	4	7	10	4	7	10
Non-defatted	0.05 ± 0.02	0.21 ± 0.02	0.29 ± 0.01	16.42 ± 19.23	28.14 ± 19.97	49.98 ± 23.40
Oil press	0.12 ± 0.01	0.22 ± 0.03	0.2 ± 0.02	7.70 ± 2.07	34.68 ± 18.60	14.54 ± 2.84
SFE	0.08 ± 0.02	0.19 ± 0.00	0.27 ± 0.02	6.04 ± 6.52	29.53 ± 16.71	62.56 ± 43.62
Combination	0.14 ± 0.03	0.23 ± 0.04	0.2 ± 0.02	16.86 ± 10.36	46.12 ± 10.94	20.48 ± 3.38

Data are presented as means ± standard deviation of two replications from two extraction batches, each (*n* = 4).

## Data Availability

The original contributions presented in this study are included in the article/Appendix A. Further inquiries can be directed to the corresponding author.

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
