# Peer review of "Effect of Defatting Method on Japanese Quince (Chaenomeles japonica) Fruit Seed Protein Isolate Technological Properties"

_foods, 2025, doi:10.3390/foods14020234_

Round 1

Reviewer 1 Report

Comments and Suggestions for Authors

The study explored three defatting methods on Japanese quince seeds and measured the properties of the defatted protein. Utilization of fruit seed has been hot topic in recent years. Overall, the study was well designed, and the results were clearly presented. Here are some tips to improve the manuscript:

1.     Title: please specify the name of the fruit seed.

2.     Keyword: please revise it to make it more proper.

3.     Line 113: for fat content determination, why don’t you choose Soxhlet extraction? How do you guarantee that all the oils have been extracted using your methods? Please provide support points and references.

4.     Table 1 and Figure 1 are replicate presentations. Please delete one. Similar to Table 4 and Figure 5, Table 5, Table 6, and Figure 6. 

Reviewer 2 Report

Comments and Suggestions for Authors

This study explores the valorization of Japanese quince seed protein and investigates its yield, purity, and functional properties using different defatting methods. The research addresses an important topic related to sustainability and the utilization of by-products. I recommend Major Revisions as this study has strong potential, but addressing the following points is necessary for further consideration.

The current title suggests a broad study applicable to all fruit seeds, while the research focuses solely on Japanese quince seeds. To better align the title with the study’s scope, I recommend specifying Japanese quince in the title. Alternatively, if you aim to generalize the findings to other fruit seeds, please justify how the results might apply more broadly. This approach would enhance clarity for the reader.

Abstract: as this research only focused on the specific seed, the opening sentence in the abstract needs to be about the Japanese quince seeds.

Introduction: Consider elaborating on why Japanese quince is significant in research or industry. This would provide a stronger foundation for the study’s relevance.

Line 87: Clarify whether seeds were milled before mixing with water for protein extraction. If they were milled, refer to them consistently as "flour" or "powder" to avoid confusion.

Line 128: Was tryptophan excluded from analysis? Because of method limitations or irrelevance to the study? You need to clarify this well. as tryptophan is an essential amino acid with a significant effect on protein functionality. If its exclusion was due to test limitations, alternative methods should be considered to include tryptophan.

Lines 248–252: Discuss potential reasons for the slight reduction in protein yield observed with SFE. Because of thermal effects?

Line 282: Elaborate on why oil press defatting yielded the highest recovery. any specific mechanisms?

Line 334: need to provide more references or detailed explanations.

Line 365: why combination-defatted protein resembles the spectra of native seeds. Could residual fat influence the spectrum? Need additional discussion.

Line 411: Can you discuss this more? Is there any with references to this statement?

Line 421: Viscosity was highest for oil press-defatted flour proteins. Do you think it was due to remaining more oil in your defatted meal?

Line 433: we need to discuss the relationship between pH, protein structure, and foaming properties. Additionally, mention applications where foaming properties at pH 10 are crucial? Is it applicable?

Lines 484–487: You state that Japanese quince protein is comparable to other nuts and seeds. Do you have references to support this claim? You need to use specific plant proteins (e.g., soy, pea) as benchmarks to provide a stronger comparison.

Lines 491–493: Higher protein yield has been observed with hydrolyzed fruit seed meals. Could you explain why hydrolysis improves yield? Additionally, discuss whether hydrolysis could be tested for Japanese quince seeds. For example, SDS-PAGE could confirm structural changes or improvements in extraction.

Polyphenolic compounds are mentioned as a limitation in the study, but their impact is not fully addressed. A brief discussion of this matter is needed.

The manuscript does not sufficiently compare the efficiency of defatting methods with existing studies.

Comments on the Quality of English Language

The manuscript is generally well-written, but there are areas where the clarity and readability can be improved. Some sentences are dense and include technical jargon that may be challenging to follow. 

Round 2

Reviewer 2 Report

Comments and Suggestions for Authors

The authors provided a thorough response to the comments.